# Correlated insulator collapse due to quantum avalanche via in-gap ladder states

Jong E. Han [1] ✉, Camille Aron[2,3], Xi Chen[1], Ishiaka Mansaray[1], Jae-Ho Han[4], Ki-Seok Kim [5], Michael Randle [6] & Jonathan P. Bird [1,6]

The significant discrepancy observed between the predicted and experimental switching fields in correlated insulators under a DC electric field far-from-equilibrium necessitates a reevaluation of current microscopic understanding. Here we show that an electron avalanche can occur in the bulk limit of such insulators at arbitrarily small electric field by introducing a generic model of electrons coupled to an inelastic medium of phonons. The quantum avalanche arises by the generation of a ladder of in-gap states, created by a multi-phonon emission process. Hot-phonons in the avalanche trigger a premature and partial collapse of the correlated gap. The phonon spectrum dictates the existence of two-stage versus single-stage switching events which we associate with charge-density-wave and Mott resistive phase transitions, respectively. The behavior of electron and phonon temperatures, as well as the temperature dependence of the threshold fields, demonstrates how a crossover between the thermal and quantum switching scenarios emerges within a unified framework of the quantum avalanche.

In spite of recent progress in our understanding of the nonequilibrium many-body state of matter, one of the long-standing problems that have remained unresolved concerns the microscopic mechanism behind the insulator-to-metal transition (IMT) of strongly correlated electronic systems, driven by a DC electric field. For more than five decades, the community has fiercely debated the origin of the dielectric breakdown in charge-density-wave (CDW) systems[1–6], for which the reported threshold electric fields are orders of magnitude smaller than theoretical estimates based on the Landau-Zener[7] mechanism. In the late seventies, this problem led to the formulation of the classical theory of depinning, in which the CDW order parameter is understood as being pinned by the presence of disorder and can be abruptly unpinned under the action of a static electric field[8–10]. More recently, a similar mismatch between theory and experiments has also been found in studies of transition-metal compounds such as Mott insulators[11–16]. In spite of the emerging potential of these

materials for applications such as non-volatile neuromorphic computing[17], the lack of understanding of the microscopic origins of their resistive transitions is a bottleneck to the development of such technologies.

From the early days of CDW research, the theoretical paradigm for the resistive transition in correlated systems has been the Landau-Zener mechanism[3,7]. This model predicts switching fields on the order of

$$E_{LZ} \sim \frac{\Delta^2}{e\hbar v_F}, \qquad (1)$$

with the (zero-field) insulating gap $\Delta$ and the Fermi-velocity $v_F$. Using typical electronic energy scales in the above expression, we obtain a rough estimate of $E_{LZ} \sim 10^{-2}$ V/Å $= 10^6$ V/cm, many orders of magnitude larger than the switching fields of $\lesssim 10$ kV/cm found in Mott insulators[11] and $1$–$10^3$ V/cm in CDW materials[1–4]. Over the years, various theoretical

[1]Department of Physics, State University of New York at Buffalo, Buffalo, NY 14260, USA. [2]Laboratoire de Physique de l'École Normale Supérieure, ENS, Université PSL, CNRS, Sorbonne Université, Université Paris Cité, F-75005 Paris, France. [3]Institute of Physics, École Polytechnique Fédérale de Lausanne (EPFL), CH-1015 Lausanne, Switzerland. [4]Center for Theoretical Physics of Complex Systems, Institute for Basic Science(IBS), Daejeon 34126, South Korea. [5]Department of Physics, POSTECH, Pohang, Gyeongbuk 37673, South Korea. [6]Department of Electrical Engineering, State University of New York at Buffalo, Buffalo, NY 14260, USA. ✉e-mail: jonghan@buffalo.edu

attempts have been made to improve the description of the resistive switching transition in a many-body context. Existing theories include the explicit time-evolution of the Hubbard chain[18], the multi-band Hubbard model[19], disorder-driven[20] and nonequilibrium phase transitions[21], spatial inhomogeneity[22], and Coulomb blockade of multi-solitons[5,23]. While bandedge broadening[21,24,25] and metal-insulator filament dynamics[22] by a uniform electric field are known to reduce the switching fields, these effects are quite modest. Recently, a nucleation mechanism for metallic domains, assisted by electron-phonon coupling, has been reported[26]. These various microscopic models have been unable to settle the aforementioned energy scale problem, or to provide clarity to the age-old debate over the role of thermal[27] versus electronic[28,29] mechanisms in the resistive transition. It has therefore long been speculated that an important ingredient must be missing.

With a wide class of switching materials[11] the mechanisms could be just as diverse. The impact ionization in semiconductors and the electro-migration in nanoscale oxide devices have been well studied. In correlated oxides much of the phenomenology has been understood in terms of the filamentary dynamics[11,12,14,27]. The recent report on the nanoscale resolution of non-filamentary patterns in $Ca_2RuO_4$[16], however, suggests quite a different non-thermal mechanism, reminiscent of the switching in organic solids[30] in which perpendicular patterns to bias have been observed. The importance of the pattern formation along the bias direction in device geometry has been pointed out by recent theories[31–33]. Given the diverse switching phenomena, we limit the scope of this paper to a new switching mechanism in the bulk correlated insulators where the switching is understood as phase transition controlled by the electric field.

The colossal mismatch of the switching fields in the field-driven Mott and CDW transitions motivates us to look for a common underlying mechanism that is shared between them. Despite the differences in CDW and Mott phenomenology, it is often hard to disentangle one mechanism from the other[34]. Here we propose that the resistive transitions in CDW and Mott insulators can be coherently explained in terms of the role of inelastic many-body processes. We show how these phonon-emitting transitions can lead to the generation of in-gap states, resulting in a quantum avalanche and a destruction of the correlated gap, at much smaller electric field scales than previously predicted.

For a conceptual understanding of the avalanche mechanism, we start by considering a one-dimensional conduction band of mass $m$ separated by a gap $\Delta$ from the Fermi energy, see Fig. 1a. The electrons are subject to three processes. First, they are accelerated by a DC electric field $E$. Second, they can emit optical phonons of energy $\hbar\omega_{ph}$

by means of an electron-phonon scattering process controlled by the coupling parameter $g_{ep}$. Finally, dephasing introduces a finite lifetime $\hbar/\Gamma$ to the electrons, setting a cutoff on the time scale over which they can form the multi-phonon state. The dephasing may arise from electron tunneling to a substrate, virtual transitions to higher electronic states, and other many-body mechanisms separate from the electron-phonon interaction. Hereafter, we adopt a unit system in which $\hbar=1$, the electric charge $e=1$, the Boltzmann constant $k_B=1$, and the lattice constant $a=1$.

Let us now give a back-of-the-envelope criterion for an avalanche that can be triggered by a combination of the three aforementioned electronic processes. We note that, in an insulator subject to an external bias, there will exist dilute, yet nonetheless finite, charge excitations. Due to the Franz-Keldysh effect[24], the bandedge smears into the gap and the transition of energetic electrons into those states by phonon emission is substantially enhanced over that in the equilibrium limit. Importantly, electron-phonon scattering processes create a ladder of intermediate replica bands that are equally spaced by $\omega_{ph}$. The minimal number of successive phonon scattering events to bridge the gap is $N_{ladder} \sim \Delta/\omega_{ph}$, see Fig. 1b. The timescale it takes to accelerate an electron in a dispersive band, to gain the energy of a single phonon, can be estimated as $\tau \sim (m\omega_{ph})^{1/2}/E$ with the electron band mass $m$. The electron-phonon scattering rate is roughly proportional to the dimensionless electron-phonon mass-renormalization factor $\lambda \equiv (g_{ep}/\omega_{ph})^2\nu_0$, where the typical density of states $\nu_0 \sim m$. The number of phonons that are emitted during the electronic lifetime $1/\Gamma$ is then $N_{ep} \sim \lambda/\Gamma\tau$. Consequently, the criterion for the avalanche becomes $N_{ep} = N_{ladder}$, yielding the following electric field required to trigger the avalanche

$$E_{av} \sim \frac{\hbar\Gamma\Delta}{eg_{ep}^2}\left[\frac{(\hbar\omega_{ph})^3}{2m}\right]^{1/2}. \tag{2}$$

This outcome is highly non-trivial in that the dissipative electron decay rate $\Gamma$ plays a crucial role in the onset of this nonequilibrium quantum phase transition. The $\Gamma \to 0$ limit is singular in nonequilibrium. By taking $\Gamma \to 0$ before or after the $E \to 0$ limit, we arrive at an effectively infinite[35] or a zero temperature (equilibrium) limit, respectively. This singular nature of the $\Gamma = 0$ limit plays a fundamental role in the avalanche, as will be confirmed below. The switching field is very different in nature from that of Eq. (1), and we note that the avalanche via inelastic scattering is also distinct from the conventional mechanism that has been discussed for high-field transport in semiconductors[36]. Most importantly, as we shall see below, the

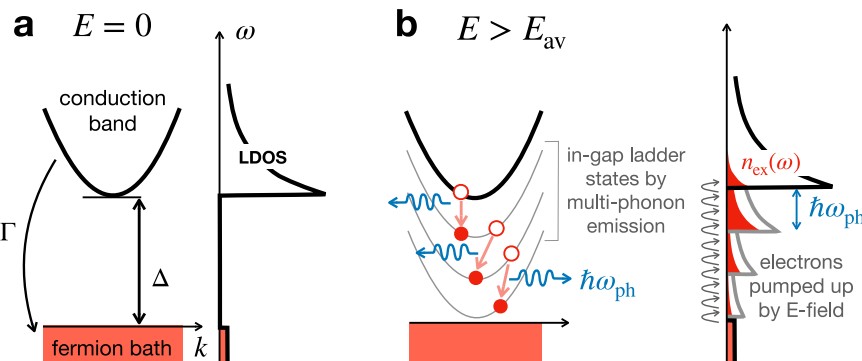

**Fig. 1 | Avalanche mechanism. a** Zero electric field, $E = 0$: (left) Spectral gap $\Delta$ between the electronic band and the Fermi level. The hybridization to a fermionic bath provides a finite decay rate $\Gamma$. (right) The local density of states (LDOS) for electrons that occupy only bath states (red shading). **b** (left) Electronic ladder-like states after successive phonon emission events (blue) form quantum pathways for

avalanche at $E > E_{av}$. (right) Once the in-gap ladder states are formed, electrons are pumped through them into the conduction band by the electric field. A finite density of electronic excitation $n_{ex}(\omega)$ at energy $\omega$ (red shading) now populates the in-gap LDOS.

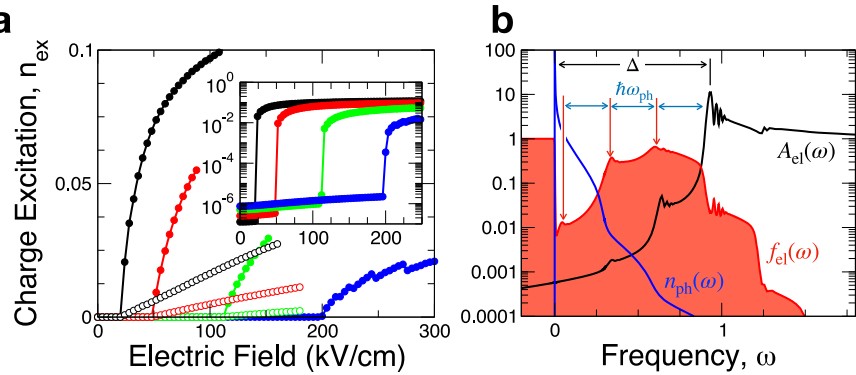

**Fig. 2 | Avalanche controlled by electron lifetime. a** Electron excitation per site $n_{ex}$ versus electric field $E$ at several values of the fermion decay rate $\Gamma = 1, 2, 4,$ $6 \times 10^{-3}$ (from left to right). The inset shows the same data on a semi-logarithmic scale. The avalanche field $E_{av}$ is proportional to $\Gamma$. The curves with filled (open) symbols are computed without (with) the nonequilibrium self-energy of the phonons. **b** Post-avalanche spectra of electrons and phonons, plotted using semi-log scales. The electron spectral function $A_{el}(\omega)$ with its bandedge at the gap $\Delta \approx 0.9$ (slightly reduced from $\Delta_0 = 1$) extends down to the reservoir level $\omega = 0$, and the distribution function $f_{el}(\omega)$ reveals in-gap state occupation with multi-phonon edges (marked by red arrows) separated by the phonon energy $\omega_{ph}$. The distribution function $n_{ph}(\omega)$ for hot-phonons is equilibrium-like.

avalanche fields predicted by Eq. (2) are much smaller than those expected from Eq. (1).

## Results

### Quantum avalanche in a rigid-band model

To demonstrate the existence of the proposed avalanche mechanism, we investigate a model of a one-band tight-binding chain under a DC electric field $E$, in the Coulomb gauge[37,38]

$$H_{0,el}^{1D} = \sum_i \left[ -t(d_{i+1}^\dagger d_i + d_i^\dagger d_{i+1}) + (2t + \Delta_0 - Ex_i)d_i^\dagger d_i \right], \quad (3)$$

where $d_i^\dagger / d_i$ is the creation/annihilation operator for an electron at site $i$, for which the site position $x_i = ia$. $\Delta_0$ is the bare gap and $t$ is the tight-binding parameter. The electrons are locally coupled to phonons, which are modeled by a collection of harmonic oscillators given by the Hamiltonian

$$H_{0,ph} = \frac{1}{2} \sum_k \left( p_k^2 + \omega_k^2 \varphi_k^2 \right), \quad (4)$$

with $\varphi_k$ the amplitude, $p_k$ the momentum, $k$ the continuum index, and $\omega_k$ the frequency of the phonon. In this section we consider the case of optical phonons, with $\omega_k = \omega_{ph}$, deferring a discussion of acoustic phonons until later. The on-site electron-phonon coupling is given by

$$H_{ep} = g_{ep} \sum_i \varphi_i d_i^\dagger d_i, \quad (5)$$

with the coupling constant $g_{ep}$.

We use the Schwinger-Keldysh formulation of the dynamical mean-field theory (DMFT)[37,39,40] which bypasses the transient dynamics and directly yields the homogeneous nonequilibrium steady-state of the many-body dynamics[41,42]. The fermion baths enter the computation of the electronic Green's function via local retarded and lesser self-energies at site $i$ as

$$\Sigma_{0,i}^R(\omega) = -i\Gamma, \quad \Sigma_{0,i}^<(\omega) = 2i\Gamma f_0(\omega + Ex_i), \quad (6)$$

while the Ohmic baths[43] enter the phonon Green's function via local self-energies[38,44] as

$$\Pi_{0,i}^R(\omega) = -2i\tau_P^{-1}\omega, \quad \Pi_{0,i}^<(\omega) = -4i\tau_P^{-1}\omega n_0(\omega). \quad (7)$$

In the above expressions, $f_0(\omega) = (e^{\omega/T} + 1)^{-1}$ and $n_0(\omega) = (e^{\omega/T} - 1)^{-1}$ are the Fermi-Dirac and Bose-Einstein distributions at the bath temperature $T$, respectively, and $\tau_P$[45] is the phonon decay time. We compute the second-order self-energy to electrons and phonons on the same footing. We refer the reader to the Methods Section and to the Supplementary Information for further details on the electron-phonon calculations.

In the calculations that follow, we choose the model parameters $\Delta_0 = 1$, $t = 1$, $g_{ep} = 0.25$, $\omega_{ph} = 0.3$, $\tau_P^{-1} = 0.001$, and $T = 0.001$ in units of eV. We use the lattice constant $a = 5$ Å to compute the electric field. We verify the schematic estimate of the avalanche field, Eq. (2), via systematic comparison with full many-body calculations over a wide range of parameters, as detailed in the Supplementary Information. There, we demonstrate the conceptual validity of the avalanche as arising from a competition between inelastic transport and dephasing and justify the limitations of Eq. (2) in the high-field limit.

Figure 2a shows the electronic excitations $n_{ex}$ per site ($0 \le n_{ex} \le 1$) when ramping up the electric field starting from the insulator state. Filled (empty) symbols denote results obtained by discarding (including) the nonequilibrium self-energy of the phonons. At the onset of the avalanche at the threshold field $E_{av}$, $n_{ex}$ increases abruptly, yet continuously, in a similar fashion as in critical phenomena. Notably, $E_{av}$ is found to be roughly proportional to $\Gamma$, consistent with the back-of-the-envelope estimate made in Eq. (2). Furthermore, the pre-avalanche excitations are inversely proportional to $E_{av}$ [see inset to (a)], which demonstrates that the avalanche is not initiated by thermal excitation but is of quantum origin. Remarkably, the onset of the avalanche is not affected by hot-phonon effects since they are only infinitesimally excited at the continuous avalanche transition, while the incoherence of phonons impedes the avalanche resulting in a linear increase of $n_{ex}$ well after the avalanche.

The non-trivial behavior of the switching field on $\Gamma$ (see Fig. 2a) should be distinguished from phonon-assisted tunneling[46,47], which is commonly manifested in resonant tunneling in heterostructures. With the electron occupation quickly reaching beyond 0.1 after the avalanche, we are in a metallic bulk limit through a phase transition, instead of in the perturbative tunneling regime. The avalanche shown here only emerges after a fully self-consistent solution is reached after hundreds of iterations, unlike what is expected from a low-order perturbative theory[46].

Figure 2b elucidates nature of the nonequilibrium state after the avalanche. The energy distribution $f_{el}(\omega)$ reveals a sizable nonequilibrium occupation of the in-gap states around energies at

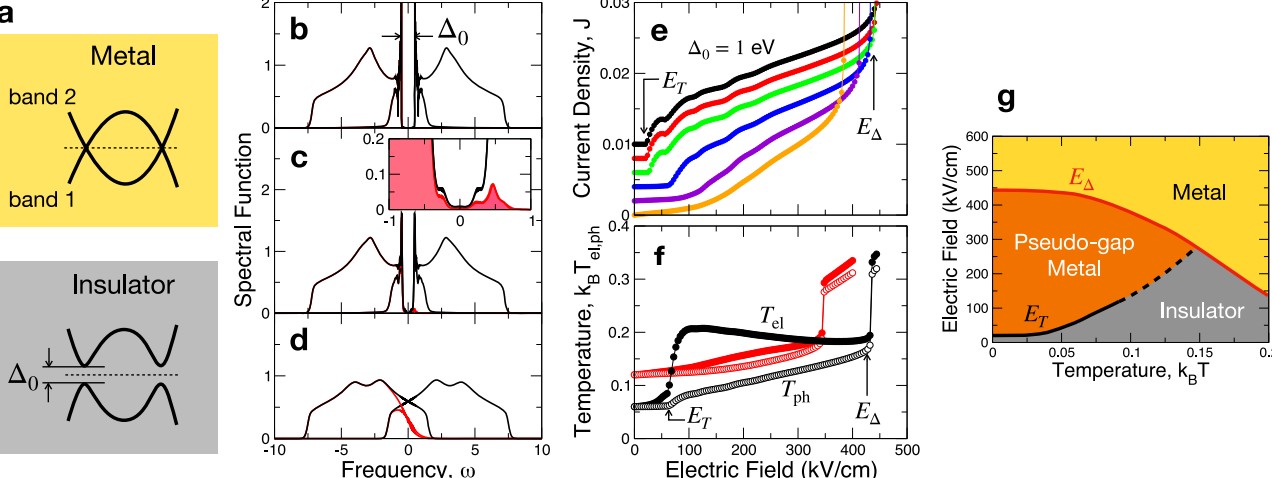

**Fig. 3 | CDW-like two-stage IMT. a** (Top) Band crossing at the Fermi level (dotted line), (Bottom) Charge gap formation due to inter-orbital interaction. **b** Density of states at electric field $E = 0$: well-defined charge gap $\Delta_0 = 1$. **c** After the first threshold $E_T$ ($E > E_T$): pseudo-gap formation. The inset is a close-up (solid red shading: the frequency-resolved electronic occupation of the in-gap states). **d** After the second threshold $E_\Delta$ ($E > E_\Delta$): metallic density of states with a fully collapsed gap. **e** Current-voltage ($I$–$V$) characteristics for various temperatures $T$ proceeds in two stages: a CDW-like continuous transition to a metal at $E = E_T$ followed by a discontinuous current jump at $E = E_\Delta$. From top to bottom, $k_B T = 0.01, 0.02, 0.04, 0.06, 0.08, 0.1$ with the curves offset in the $y$-axis by 0.002 for readability. **f** Effective temperatures $T_{el,ph}$ of the electrons (filled circle) and phonons (empty circle), respectively. The bath temperatures are $k_B T = 0.06$ (black) and 0.12 (red). At $E = E_T$, the electrons go through a transition without heating up the phonons, while at $E_\Delta$ the two temperatures equilibrate. **g** Phase diagram in the $E$–$T$ plane. The $E_T$ and $E_\Delta$ curves delimit a pseudo-gap metal region. Dashed line denotes the high temperature limit where the phase boundary is smeared. (See the main text for the parameters.).

multiples of $\omega_{ph}$ away from the bandedge. This confirms the involvement of multi-phonon emission in the avalanche mechanism. Moreover, the strong deviation of $f_{el}(\omega)$ from a Fermi-Dirac distribution points to the non-thermal nature of the avalanche. In contrast, the phonon distribution $n_{ph}(\omega)$ appears mostly thermal.

**Insulator-to-metal transition induced by avalanche**

Having established the existence of the avalanche, we now turn our attention to the implications of the avalanche in the context of the resistive transition in correlated insulators. The strong charge fluctuations initiated by the avalanche are expected to generate phonon excitations, which then profoundly perturb the inter-orbital mixing and destabilize the charge gap. To address this point, we extend our model to a two-band correlated insulator where the gap $\Delta$ between the bands is generated by electronic interactions. (See Fig. 3a) Specifically, we consider the following Hamiltonian which allows us to address both the CDW transitions and resistive switching on an equal footing,

$$
\begin{aligned}
H_{el}^{2D} = & -t \sum_{\langle ij \rangle} \sum_{\alpha=1,2} (-1)^\alpha (d_{\alpha i}^\dagger d_{\alpha j} + d_{\alpha j}^\dagger d_{\alpha i}) \\
& + \sum_i \sum_\alpha [(-1)^\alpha (2t - \mu) - E x_i] d_{\alpha i}^\dagger d_{\alpha i} \\
& + \sum_i \left[ \xi(d_{1i}^\dagger d_{2i} + d_{2i}^\dagger d_{1i}) + \frac{\xi^2}{2U} \right].
\end{aligned}
\tag{8}
$$

Here, $\alpha$ is the band index and $i$ is the site index on a square lattice with $\langle ij \rangle$ denoting nearest neighbors. The last term in Eq. (8) is the decoupling of an inter-orbital Hubbard-like interaction via an auxiliary quantum field $\xi$. The magnitude of $\xi$ respresents the order parameter that opens a gap, i.e. the strength of the density modulation in CDW or the charge fluctuations in correlated insulators. The phase of $\xi$ captures the phase slip in CDW materials or the Goldstone excitations in correlated insulators. In this work, we investigate the instability of a uniform symmetry-broken state, and ignore the phase fluctuations of $\xi$ for simplicity. Consideration of $\xi$-fluctuations would only weaken the order parameter and lead to an even earlier collapse of the insulating state, and thus further supporting our conclusions. The static mean-field condition becomes

$$
\xi = U \langle d_{1i}^\dagger d_{2i} + d_{2i}^\dagger d_{1i} \rangle.
\tag{9}
$$

The electron-phonon coupling is given by $H_{ep} = g_{ep} \sum_{i,\alpha} \varphi_{\alpha i} d_{\alpha i}^\dagger d_{\alpha i}$, with the independent phonons coupling to each orbital. The fermion/phonon baths are set up as described previously with the Fermi energy at the band crossing.

The spectrum of the phonons has important consequences for the IMT. Let us first discuss the electric field-driven IMT in the presence of optical phonons with energy $\omega_{ph}$, which we will associate with the multi-stage transitions that are commonly observed in CDW systems. Later, we will turn to the case of acoustic phonons which we will relate to Mott systems. With increasing electric field, the insulating system undergoes a two-stage transition to a metal as manifested by the spectral function in Fig. 3b–d. Here, we adjust the electron-phonon coupling $g_{ep}$ such that, given $U = 2$, the initial gap $\Delta_0$ is tuned to 1.0 at $E = 0$ and at the lowest temperature. The strength of this coupling corresponds to a moderate (dimensionless) mass-renormalization factor of $\lambda \approx (g_{ep}/\omega_{ph})^2 \nu_0 = 0.32$ with the phonon energy $\omega_{ph} = 0.3$ eV and the 2D density of states $\nu_0 \approx (8t)^{-1}$. (The damping is set to $\Gamma = \tau_P^{-1} = 0.001$ eV[48], and the chemical potential to $\mu = t$.) In the low-field limit, the spectrum features a well-defined charge gap. As $E$ increases beyond the first threshold field $E_T$, in-gap states develop via the avalanche mechanism. This results in a pseudo-gap phase where the system becomes metallic while the charge gap mostly remains intact. The energy-resolved electron occupation $n_{ex}(\omega) = (2\pi)^{-1} \mathrm{Im} G^<(x_i = 0, \omega)$ (the red shaded area shown in the inset to panel **c**) indicates that the electric current is carried mainly by the states of the conduction band above the gap while the in-gap states provide the pathway for this population inversion. The pseudo-gap regime persists until $E = E_\Delta$, when the gap $\Delta$ collapses to zero in a strongly discontinuous transition.

The current-voltage ($I$–$V$) relation in Fig. 3e shows evidence of a two-stage IMT. First, the system continuously becomes metallic at the threshold field $E = E_T$. Later, at the higher threshold $E_\Delta$, the current rises discontinuously. The avalanche behavior discussed in Fig. 2 is responsible for the threshold behavior at $E_T$. This behavior is similar to

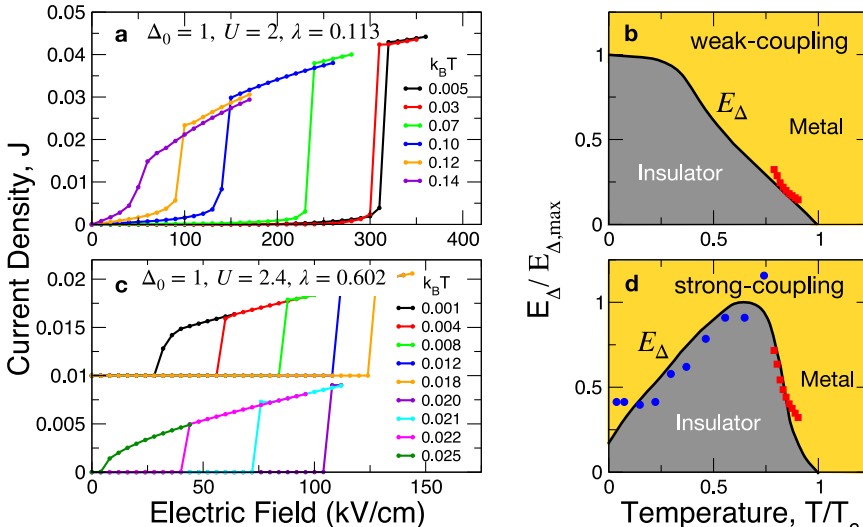

**Fig. 4 | Mott-like abrupt IMT. a** Current-voltage ($I–V$) relation in the limit of weak coupling to acoustic phonons, displaying a single-stage discontinuous insulator-to-metal transition (IMT). **b** Switching field versus temperature ($E_\Delta$ vs. $T$) phase diagram for **a** normalized to the maximum of $E_\Delta$ and the transition temperature $T_c$, respectively. $k_B T_c = 0.15$. The red squares are switching fields of $W_x V_{1-x} O_2$ with $x = 0.0114$ and $T_c = 280$ K (adapted from ref. 55, y-axis arbitrarily scaled). **c** $I–V$ relation in the strong-coupling limit showing non-monotonic dependence of $E_\Delta$ on $T$. **d** Non-monotonic $E_\Delta$ versus $T$. The initial increase of $E_\Delta$ at low $T$ demonstrates the non-thermal nature of the IMT in the strong-coupling limit. $k_B T_c = 0.025$. The blue circles are switching fields of $TiS_3$ with $T_c = 290$ K (adapted from refs. 56,57, y axis arbitrarily scaled). The red squres are the same as **b**.

that in CDW systems, in which it has been widely attributed to the depinning transition[8,9]. Here, we propose an alternative mechanism of electron avalanche via inelastic scattering that does not require any disorder or reduced dimensionality. With coupling to optical phonons, the avalanche is not sufficiently disruptive to the gap, and the intermediate gapped state is instead sustained over a wide range of electric field ($E_T < E < E_\Delta$). We note that $E_T$ is around two orders of magnitude smaller than the switching field expected for zero electron-phonon coupling strength $E_\Delta(\lambda = 0, \Delta_0 = 1) \approx 1.6$ MV/cm.

The non-thermal nature of the avalanche transition is illustrated by the electric field dependence of the effective temperatures for electrons and phonons, as shown in Fig. 3f. (See Methods for the definition of effective temperatures.) As the electric field is increased beyond $E = E_T$, the electrons heat up immediately while the phonons stay cold. This clearly demonstrates that heat exchange does not trigger the avalanche. On the other hand, the full IMT at $E_\Delta$ is initiated once the electron and phonon temperatures equilibrate, suggesting that this second transition can be described in terms of a thermal mechanism. It is remarkable to demonstrate that the mechanisms for electronic and thermal switching, the topic of intense discussions in the literature, are not exclusive of each other but can arise simultaneously from a single microscopic model.

The phase diagram defined by the two switching fields ($E_T$ and $E_\Delta$) is plotted in Fig. 3g. The most notable observation here is that $E_T$ remains constant near zero temperature, but increases at higher temperatures until it merges with $E_\Delta$. This may seem counter-intuitive, since the gap is expected to decrease with increasing $T$. The gap, however, remains nearly constant until close to a critical temperature $T_c$, meaning that a thermal argument is not applicable. We find that, as suggested by the hot-phonon effects apparent in Fig. 2, thermal decoherence may be detrimental to the avalanche, meaning that a stronger electric field is required to induce an avalanche. This is a further evidence that the threshold behavior is of quantum-mechanical origin. In contrast to $E_T$, $E_\Delta$ decreases with $T$, and is conventional and thermally driven.

The existence of bias-driven multiple-stage transitions in CDW systems has been intensely debated[1,49–52]. For instance, recent ARPES studies on the $NbSe_3$ system have shown unambiguously that the various CDW gap energies are constant (marginally increasing) with

increasing temperature and observable even beyond their respective $T_c$ values[53,54]. This suggests a non-thermal mechanism of gap formation below $T_c$. At temperatures beyond $T_c$, the gap gradually closes suggesting a thermal mechanism, where these predictions show an interesting parallel to our results. What our model as a conceptual framework showed are that the nature of the initial threshold[51] in the low-field limit is quantum in the sub-gap energy scale, and that, after an intermediate pseudo-gapped phase, there are subsequent discontinuous resistive transitions that thermally destroy the order parameter.

Finally, let us discuss the case of electron coupling to acoustic phonons, with a continuous spectrum of the form $\omega_k \propto k$ and cutoff Debye energy chosen as $\omega_D = 0.6$ eV. Compared to optical phonons, the influence of acoustic phonons on the nonequilibrium dynamics is more dramatic. We discuss this situation for both the weak and strong limits of electron-phonon coupling, as shown in Fig. 4. In the weak-coupling limit ($\lambda = 0.113$), corresponding to panels (a) and (b), the IMT is dominated by a strongly discontinuous collapse of the gap. While the signature of the avalanche is still present as a precursor to the IMT, the range of the avalanche region is much narrower than that found for coupling to optical phonons. The avalanche current is also very small so that its effect is insignificant. $E_\Delta$ decreases monotonically with increasing temperature and the $E–T$ phase diagram is conventional and fully consistent with the thermal switching scenario. We tested this result against experimental data by overlaying the experimental switching fields in the Vanadium oxides $W_x V_{1-x} O_2$[55] at temperatures close to the transition temperature. In addition to an overall agreement of the trends, the lightly concave curve shape[55,58] found in the experiment is reproduced by the theory.

In the strong electron-phonon coupling limit ($\lambda = 0.602$), on the other hand, $E_\Delta$ is strongly non-monotonic, increasing with $T$ in the low-$T$ limit. As is especially apparent from panel (c), the intermediate field region between $E_T$ and $E_\Delta$ has diminished dramatically so that the IMT appears as a fully single-stage transition, as often observed in correlated transition-metal oxides[11,12,27,59,60]. In this limit, the strong low-energy excitations of acoustic phonons cause the IMT to bypass the CDW-like pseudo-gap state. We therefore make a prediction that the avalanche physics which controls the CDW threshold field $E_T$ is manifested in Mott insulators in a switching field $E_\Delta(T)$ that increases with

*T*. We compare our results with the single-stage switching fields measured in TiS$_3$[56,57] (blue circles). The agreement between theory and experiment is quite reasonable. Altogether, the $E_\Delta(T)$ behavior in the weak- and strong-coupling limits demonstrates a crossover between the quantum and thermal switching mechanisms as temperature is varied.

To conclude, we have established a quantum avalanche mechanism as a generic scenario for the nonequilibrium phase transition in correlated insulators. The proposed mechanism not only resolves the long-standing discrepancy between the observed and predicted switching fields in these materials, but also sheds new light on the origins of the crossover between the quantum and thermal scenarios in resistive transitions. The existence of the pseudo-gapped metallic states may be directly verified experimentally by using transient electrical pulses of varying duration to control the amount of hot-phonon generation. While we have discussed this mechanism with electron-phonon coupling, the avalanche may also arise via coupling to other bosonic excitations, such as the Goldstone modes associated with the order parameter responsible for the opening of the charge gap. Such a scenario may provide a more direct path for the destruction of the correlated gap. Here we have presented a minimal framework for the quantum avalanche that is fundamentally different from the Landau-Zener mechanism. The study of the avalanche with spatial inhomogeneity, and its interplay with disorder, is left for future research.

## Methods

### Calculation of self-energy and effective temperature

The many-body calculations in this work are based on the nonequilibrium Green's function technique approximated by the DMFT scheme[37,38,40,44], in which we limit the self-energy to be diagonal in site and orbital indices. Full details on the nonequilibrium DMFT are given in the Supplementary Information. The electron and phonon self-energies, expressed respectively as

$$\Sigma^{\lessgtr}_{ep,\alpha}(\mathbf{r},\omega) = ig^2_{ep} \int \frac{d\omega'}{2\pi} \mathcal{G}^{\lessgtr}_{\alpha\alpha}(\mathbf{r},\omega-\omega')D^{\lessgtr}_{\bar{\alpha}}(\omega'), \tag{10}$$

$$\Pi^{\lessgtr}_{ep,\alpha}(\omega) = -2ig^2_{ep} \int \frac{d\omega'}{2\pi} G^{\lessgtr}_{\alpha\alpha}(\mathbf{r},\omega+\omega')G^{\gtrless}_{\alpha\alpha}(\mathbf{r},\omega'), \tag{11}$$

are iterated to convergence. (The factor 2 in the phonon self-energy accounts for spin degeneracy.) The self-energies without the vertex correction, i.e., Migdal approximation[61], are reasonable since we are in the weak- to moderate el-ph coupling limit, as can be seen with less than 10% level shift of the bandedge in Fig. 2b.

Once we achieve full convergence, we compute the distribution functions as

$$f_{el}(\omega) = -\frac{1}{2}\frac{\sum_\alpha \mathrm{Im}G^<_{\alpha\alpha}(\mathbf{r}=0,\omega)}{\sum_\alpha \mathrm{Im}G^R_{\alpha\alpha}(\mathbf{r}=0,\omega)}, \quad n_{ph}(\omega) = \frac{1}{2}\frac{\mathrm{Im}D^<(\omega)}{\mathrm{Im}D^R(\omega)}. \tag{12}$$

We define the effective temperature for electrons and phonons in terms of the first moment of the distribution, which correctly reduces to the bath temperature in the equilibrium limit, as

$$T^2_{el} = \frac{6}{\pi^2}\int_{-\infty}^{\infty} \omega[f_{el}(\omega) - \Theta(-\omega)]d\omega$$
$$T^2_{ph} = \frac{6}{\pi^2}\int_0^\infty \omega n_{ph}(\omega)d\omega, \tag{13}$$

with the step-function $\Theta(x)$. As shown in Fig. 2b, electronic distribution functions may deviate strongly from the Fermi-Dirac form, in which cases $T_{el}$ provides an approximate measure of nonequilibrium energy excitation.

## Data availability
The data that support the findings of this study are available from the first author. Source data are provided with this paper.

## Code availability
The computational code required to reproduce the theoretical calculations is available from the corresponding author upon reasonable request.

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

## Acknowledgements

J.E.H. is grateful for computational support from the CCR at Buffalo. J.E.H. is supported by Air Force Office of Scientific Research under award no. FA9550-22-1-0349. CA acknowledges the support from the French ANR "MoMA" project ANR-19-CE30-0020 and from the project 6004-1 of the Indo-French Center for the Promotion of Advanced Research (IFCPAR). J.H.H. acknowledges the support from the Institute for Basic Science in the Republic of Korea through the project IBS-R024-D1. KSK was supported by the Ministry of Education, Science, and Technology (Grants No. NRF-2021R1A2C1006453 and No. NRF-2021R1A4A3029839) of the National Research Foundation of Korea (NRF) and by TJ Park Science Fellowship of the POSCO TJ Park Foundation. We are much grateful to Sambandamurthy Ganapathy, Han-Woong Yeom, Emmanuel Baudin for their helpful discussions.

## Author contributions

J.E.H. conceived the project and produced the theoretical data. C.A., X.C., and I.M. helped the development of the code. C.A., X.C., I.M., J.H.H., and K.S.K. contributed to theoretical discussions. M.R. and J.P.B. contributed with experimental discussions and with the interpretation of the theoretical results. The manuscript was written by J.E.H., C.A., M.R., and J.P.B. with contributions from all authors.

## Competing interests

The authors declare no competing interests.
