## [Peer Review File · Nature Communications]

REVIEWER COMMENTS

Reviewer #1 (Remarks to the Author):

The authors proposed a generic model of electrons coupled to an inelastic medium of phonons, and demonstrated that an electron avalanche can occur in the bulk limit of such insulators at arbitrarily small electric field. A quantum-avalanche mechanism can reproduce the nonequilibrium IMT in gapped system.

Since this manuscript provides interesting results and the conclusions of the paper are supported by the well-established theories and models, I recommend this manuscript for publication. However, there are some points that need to be considered, see below:

- (1) How to judge the validity of the employing parameters to reproduce Figs. 2-4?
- (2) The fermion/phonon baths are set up as described previously with the Fermi energy in the center of the gap, which allows to address both the CDW Peierls transition and Mott physics on an equal footing. The authors should mention the correspondence with the experimental results.
- (3) Address the parameter (index) values in Eqs. (8) and (9) and explain the validity.

Reviewer #2 (Remarks to the Author):

The paper by Han et al. proposes a new approach to the field driven metallisation of charge density wave (CDW) and Mott insulators. They suggest an interesting physical picture, where the presence of a potential gradient accelerates electrons, these electrons lead to multiple optical phonon emission (via electron-phonon coupling), the phonons create electronic subbands (mid gap spectra in the insulator) and the pile up of such weight leads to current flow well before the Landau-Zener (LZ) threshold is reached. Their avalanche field, obtained by a back of the envelop estimate, is

$$E_{av} \sim (\Gamma \Delta / g^2_{ep}) \Omega_{ph}^{3/2},$$

where Δ is the zero field gap, Γ is the electron damping

rate, g_{ep} is the electron-phonon coupling, and Ω_{ph} is the optical phonon frequency.

This is in contrast to the LZ estimate $E_c \sim \Delta^2/v_F$

The proposed result not only differs from LZ in terms of its scaling with Δ , but also the introduction of (and emphasis on) multiple phonon parameters and a seemingly "extrinsic" damping scale Γ (which cannot be calculated within the present scheme but is taken as an input). Even in seemingly 'non-phonon' systems like Mott insulators, the authors would suggest that phonons play a crucial role in real life transitions.

This is an innovative idea in an otherwise crowded and technically evolved field. It would be of value if (i) it is borne out by a credible many body calculation, and (ii) the predictions compare successfully with existing data. In my reading of the present version of the paper neither seems to be achieved.

Let me expand on (i). The field of nonequilibrium correlated systems is hard because correlated systems are difficult to solve even at equilibrium. One would trust the nonequilibrium results provided the method has the capacity to reproduce non trivial physics at equilibrium. I start on this aspect before moving to the field driven situation.

1. The authors investigate a 1D electron-phonon problem in eqn.3. to highlight the avalanche issue.

This model at half filling would lead to a state with CDW

correlations, and - in general - at strong coupling would lead

to polaronic quasiparticles. Quite naturally the technical details are not presented in the main paper but an examination of the Suppl reveals that the Keldysh scheme primarily uses an one loop self consistent method for the electron and phonon self energies.

This is akin to the Migdal scheme at equilibrium (now including the potential gradient) and hardly adequate to capture the strong coupling physics of electron-phonon systems. If the authors never intended to approach strong coupling physics that is not clearly stated. And if they indeed remain at weak coupling the E_{av} suppression in eqn.2 is not going to be significant.

2. Maybe I am missing something, but there would be a density modulation in the CDW phase, I cannot see where/if that is included in eqn.3, unless the authors do not consider half filling or the CDW instability is simply ignored.

3. Eqn.8 defines a more general model for Mott/CDW insulators.

This involves a mean field version of the Hubbard interaction (encoded by a parameter ξ) and phonon physics as already presented in eqn.3. If one were to switch off the electron-phonon coupling this should be the model for a Mott insulator.

Is it? Even within DMFT there is a sophisticated theory of a Mott transition - and I am not sure a ξ based theory gets that right. Once ξ vanishes the model becomes a band problem - with no correlation effects - unlike what would happen in a real Mott insulator. Like in the case of the electron-phonon

aspect I am not sure the approach adopted gets much of the equilibrium Mott/correlated metal physics right.

4. What about the finite field case? Apart from shortcomings in handling the many body aspect, the assumption of a linear potential in the system is an oversimplification. The effect of a potential difference - introduced via leads - has been studied carefully and leads to a non trivial potential profile in the system, Ribero et al. Phys. Rev. B. 93, 144305 (2016), Tanaka et al., Phys. Rev. B 83, 085113 (2011), Dutta et al., Phys Rev B 101, 245155 (2020). None of these seem to be known to the authors. As a result the entire spatial aspect of the breakdown process is lost in the present paper. This spatial aspect is not incidental, it suggests that the insulator-metal transition starts at the edge.

Now on to (ii).

5. Only recently experimenters have started spatial mapping of the electronic state near a bias driven transition, see, e.g, Zhang et al., Phys Rev X 9, 011032 (2019). In Ca₂RuO₄ it shows via near field optical microscopy how the effect of a voltage bias propagates into the system. The bias driven metallisation

process is not homogeneous.

6. The temperature dependence of the critical voltage (or field) in a wide variety of materials shows E_c reducing with increasing temperature. This ranges from the oxides - Ca₃Ru₂O₇ and VO₂ - to the organics kappa-BEDT and K-TCNQ. If the avalanche

mechanism in Fig.3(g) were at play that would not have been the case. In fact in a correlated insulator the gap itself would reduce with T - weakening the threshold field - so it is hard to see where something like 3(g) is relevant.

7. Apart from highlighting a scenario I am unable to see what existing discrepancy is actually explained, and data organised, by the authors. Given the note in which the introduction is written one would have hoped for an actual comparison of the predictions of the "avalanche approach" with breakdown fields and their temperature dependence in several materials.

Let me summarise:

The authors have introduced an interesting scenario that could play a role in bias/field driven transition in insulators. Electron-phonon coupling and a somewhat mysterious dephasing rate play a crucial role in the process - involving multiphonon emission and emergence of subgap states. This much is a conceptual advance and the effect could well play a role in real materials. However the treatment of actual correlated models is rather naive, some assumptions and predictions are clearly not consistent with experiments, and no effort is made to test out the predictions (whatever the methodological limits) against real data.

I think the interesting idea should have a stronger foundation in theory and there should be an attempt at a serious comparison with experiments to merit publication in Nat Comm.

We thank the Reviewers for their thoughtful comments that we have taken seriously to improve our manuscript. Both reviewers acknowledged the novelty of the quantum avalanche mechanism for the resistive transition. Reviewer #1 found the work interesting and recommended it for publication. Reviewer #2 considered the work as "innovative", "an interesting idea" and "a conceptual advance". However, both referees questioned and required clarification for the parameters chosen in this work, and more importantly, explicit comparison to experiments to prove the validity of this work.

Before we make itemized replies to each criticism, let us begin by giving a brief summary of revisions made in the manuscript.

We agree with the Reviewers that the goals of this work have not been clearly presented, and, therefore, revised the introduction and the conclusion with more explicit statements. The three main goals have been (i) to resolve the huge energy-scale mismatch for the switching electric field, (ii) to shed light on the age-old debate between the electronic vs thermal mechanism for the resistive transition, and (iii) to combine the seemingly different resistive switching systems of CDW and Mott insulators within a single model. We accomplished the goals by introducing the quantum avalanche mechanism through multi-phonon emission.

In addition to the introduction and conclusion sections, changes have been made to Fig.2, 3, and 4. In particular, we added experimental results to support our theoretical findings in Fig.4. As stated categorically below in our reply to Reviewer #2 in item 6, we show that the theory is consistent with (i) the threshold field behavior in CDW at low temperature limit, in agreement with well-established CDW literature, (ii) the high temperature behavior of the IMT in Mott systems, compared with an experimental IMT from VO₂ close to the transition temperature. We make a prediction (iii) in the low temperature limit of the IMT in Mott systems and we make a comparison with a recent experimental IMT from TiS₃.

Motivated by Reviewer #2 comments 4 and 5 on the inhomogeneity in the resistive switching as observed in Ca₂RuO₄, we added a new paragraph in which we discussed different aspects of switching mechanisms, and defined the scope of our work.

We added a section to Supplementary Material to address the concerns of both Reviewers on the choice of model parameters. The most crucial is the phonon frequency, and we have demonstrated the robustness of the avalanche there.

The original manuscript was transferred directly from Nature Physics only with minor format changes, and some discussions were not sufficient. As pointed out by Reviewers, we added more detailed discussions for the justifications and model parameters in the revised version. We have added all references mentioned in the Reviews.

Minor changes:

(1) In Fig.2(b), we used a semi-log plot to show the in-gap tail of the

spectral function.

(2) We changed E_{IMT} to E_{Δ} . E_{IMT} is the standard notation for the switching in Mott systems, where the switching is single-stage. However, while the gap-vanishing transition in the CDW system is the second threshold, the metallization occurs at the first transition which we labeled as E_T . Therefore, characterizing the second transition in CDW by E_{IMT} is misleading and we have renamed all E_{IMT} as E_{Δ} , denoting the gap-vanishing transition. Accordingly, Fig.3 and 4 were revised.

>
> Reviewer #1 (Remarks to the Author):
>
> The authors proposed a generic model of electrons coupled to an
> inelastic medium of phonons, and demonstrated that an electron
> avalanche can occur in the bulk limit of such insulators at
> arbitrarily small electric field. A quantum-avalanche mechanism can
> reproduce the nonequilibrium IMT in gapped system.
> Since this manuscript provides interesting results and the
> conclusions of the paper are supported by the well-established
> theories and models, I recommend this manuscript for publication.
> However, there are some points that need to be considered, see below:
>
> (1) How to judge the validity of the employing parameters to reproduce
Figs. 2-4?
>

As with typical lattice models, we used the tight-binding parameter and the gap at the generic value 1 eV. The crucial parameters are phonons and their coupling constant. As pointed out by Reviewer #2 as well, the concern is if we used too strong an el-ph constant to exaggerate the avalanche. In Fig.2, the effect of the el-ph coupling can be assessed by the shift of electronic levels. As can be seen in FIG.2(b), the bare bandedge at 1.0 is shifted downward by less than 0.1, which is a fairly weak effect. Furthermore, as can be seen in FIG. 2(a), the inclusion of the phonon self-energy does not affect the onset of the avalanche at all (since the transition is a continuous phase transition). Therefore we are in a generic and weak-coupling regime. The same can be said for FIG.3 and FIG.4(a-b), and the gap at zero field is well defined at 1, without any polaronic effects. For FIG.4(c-d), the coupling constant is raised to give the dimensionless coupling constant $\lambda=0.602$ (defined with the ungapped DOS). Not only this value is what one often finds in the literature, but also the effectiveness of the el-ph coupling is much reduced in the gapped limit. Therefore, we do not think we fine-tuned the parameters to maximize the avalanche effect.

With the el-ph coupling in the weak-to-moderate range, we also studied the avalanche fields as a function of the phonon energy w_{ph} while keeping the coupling energy $(g_{ep}/w_{ph})^2$ fixed at $(0.25/0.3)^2$ as given in FIG2. The results are

```
=====
wph      E_av
-----
0.3      0.0010 (same as FIG2, in unit of the gap)
```

0.2 0.00095
0.1 0.00086
0.05 0.00082 (with estimate error of 0.00002, E-field step size)

This shows that the theory does not require any unrealistically large values for the phonon energy, and the avalanche is extremely robust. We added a section discussing this result in the Supplementary Information (SI). We mentioned this confirmation at the end of the first paragraph on page 6 of the main text.

For the dissipation rate Γ , it is hard to give a first-principles estimate. One knows for sure that, with a large line broadening by Γ , even an equilibrium phase transition would be smeared. The equilibrium transitions are typically quite sharp and we, therefore, expect Γ to be much smaller than typical electronic energy scales. Diaz, Aron and Han (two of us) recently worked out an estimate of Γ based on the experimental dissipation power as being less than 1 meV, squarely in the range of Γ we used in this work. We cited the reference as [47].

For further discussions on parameters, please see replies to Reviewer #2.

>
> (2) The fermion/phonon baths are set up as described previously with
> the Fermi energy in the center of the gap, which allows to address
> both the CDW Peierls transition and Mott physics on an equal footing.
> The authors should mention the correspondence with the experimental
> results.
>

Reviewer #2 also raised the same concerns for experimental comparison categorically in his/her comments 5-7, where our detailed replies can be found. To summarize, we categorized the experimental facts into three parts where the first two, experimentally well-established, are in agreement with our model, and the third item is our prediction which needs future experimental verification. Revised Fig.4 addressed the experimental comparison.

>
> (3) Address the parameter (index) values in Eqs. (8) and (9) and explain
> the validity.
>

Please see the reply to (1) above and the comments below.

>
> Reviewer #2 (Remarks to the Author):
>
>
> The paper by Han et al. proposes a new approach to the field driven
> metallisation of charge density wave (CDW) and Mott insulators. They
> suggest an interesting physical picture, where the presence of a
> potential gradient accelerates electrons, these electrons lead to
> multiple optical phonon emission (via electron-phonon coupling), the
> phonons create electronic subbands (mid gap spectra in the insulator)
> and the pile up of such weight leads to current flow well before the

> Landau-Zener (LZ) threshold is reached. Their avalanche field, obtained by a back of the envelop estimate, is $E_{av} \sim (\Gamma \Delta / g_{ep}^2) \Omega_{ph}^{3/2}$, where Δ is the zero field gap, Γ is the electron damping rate, g_{ep} is the electron-phonon coupling, and Ω_{ph} is the optical phonon frequency.

> This is in contrast to the LZ estimate $E_c \sim \Delta^2 / v_F$

> The proposed result not only differs from LZ in terms of its scaling with Δ , but also the introduction of (and emphasis on) multiple phonon parameters and a seemingly "extrinsic" damping scale Γ (which cannot be calculated within the present scheme but is taken as an input). Even in seemingly 'non-phonon' systems like Mott insulators, the authors would suggest that phonons play a crucial role in real life transitions.

> This is an innovative idea in an otherwise crowded and technically evolved field. It would be of value if (i) it is borne out by a credible many body calculation, and (ii) the predictions compare successfully with existing data. In my reading of the present version of the paper neither seems to be achieved.

> Let me expand on (i). The field of nonequilibrium correlated systems is hard because correlated systems are difficult to solve even at equilibrium. One would trust the nonequilibrium results provided the method has the capacity to reproduce non trivial physics at equilibrium. I start on this aspect before moving to the field driven situation.

> 1. The authors investigate a 1D electron-phonon problem in eqn.3. to highlight the avalanche issue.

> This model at half filling would lead to a state with CDW correlations, and - in general - at strong coupling would lead to polaronic quasiparticles. Quite naturally the technical details are not presented in the main paper but an examination of the Suppl reveals that the Keldysh scheme primarily uses an one loop self consistent method for the electron and phonon self energies. This is akin to the Migdal scheme at equilibrium (now including the potential gradient) and hardly adequate to capture the strong coupling physics of electron-phonon systems. If the authors never intended to approach strong coupling physics that is not clearly stated. And if they indeed remain at weak coupling the E_{av} suppression in eqn.2 is not going to be significant.

We appreciate the Reviewer for pointing out the approximations we employed in the theory. We stated the Migdal approximation explicitly in the manuscript in the Method Section below Eq.(11). In this work, however, we are working in the weak coupling regime as noted in the reply to Reviewer #1, and particularly because the Fermi energy is outside the band, we are in the zero-filling limit for the model, Eq.(3). Even though we normalized the el-ph coupling constant by the tight-binding parameter to give a dimensionless λ for a figure of merit the el-ph coupling, the relevant DOS at Fermi energy is zero. The

weak el-ph coupling can be visualized in FIG2(b) where the band edge (after an avalanche) remains well separated from the Fermi energy. (The el-ph binding energy is less than 1/10 of the gap.) Therefore we are far from the conditions needed for the polaron physics in any regime we explored. Regarding Eq.(2), the expression was given for a back-of-the-envelope estimate, and with the small value of Γ we reach the numerically demonstrated E_{av} at 0.001 of the gap, which is unprecedentedly small for a theoretical estimate in a bulk limit model.

Regarding the "extrinsic" comment, we understand the Reviewer mentioned this in terms of fermion against phonon baths. In the pre-avalanche limit, the current is very small and the phonons are not yet activated. Nonetheless, there exists a finite current (the insulating resistivity is not infinite) and it dissipates via other channels which we represent by the fermion baths that provide the electron lifetime in the low-energy range. We leave it open for debate whether the fermion baths are "extrinsic". However, the zero-dissipation limit of $\Gamma=0$ is an unstable nonequilibrium fixed point, the conceptual ramification of which should be debated and explored in future theories. We feel strongly that the theoretical community recognizes the importance of the nonequilibrium fixed point that we added a discussion on this below Eq.(2).

Additional discussions on the model parameters can be found in the reply to Reviewer #1, which we briefly summarize here. We investigated the avalanche for lower phonon energies and found the avalanche remained quite robust. (We added a section to SI to address this) Furthermore, we give an empirical justification for the damping parameter Γ .

>
> 2. Maybe I am missing something, but there would be a density
> modulation in the CDW phase, I cannot see where/if that is
> included in eqn.3, unless the authors do not consider half
> filling or the CDW instability is simply ignored.
>

We thank the Reviewer for pointing out the important aspect of the model. The CDW phase shift within the coarse-grained model of Eq.(8) will appear as the phase factor to the complex order parameter ξ . As will be elaborated further below (to comment 4), we limited our analysis to the instability of a uniformly ordered phase and the CDW phase would not explicitly appear within our formulation. We discuss this aspect of the CDW phase fluctuation in the text just below Eq.(8).

> 3. Eqn.8 defines a more general model for Mott/CDW insulators.
> This involves a mean field version of the Hubbard interaction
> (encoded by a parameter ξ) and phonon physics as already
> presented in eqn.3. If one were to switch off the electron
> -phonon coupling this should be the model for a Mott insulator.
> Is it? Even within DMFT there is a sophisticated theory of a
> Mott transition - and I am not sure a ξ based theory gets
> that right. Once ξ vanishes the model becomes a band problem
> - with no correlation effects - unlike what would happen in a
> real Mott insulator. Like in the case of the electron-phonon
> aspect I am not sure the approach adopted gets much of the

> equilibrium Mott/correlated metal physics right.
>

We understand the Reviewer's concerns as to whether our mean-field treatment of the Mott physics is adequate, and also fully agree that a full many-body treatment will modify the results. We want to make clear, however, that our goal is to investigate how the long-range order can be destabilized by the charge fluctuations while settling the issue of the energy-scale discrepancy. Therefore, while we agree that our $\xi=0$ solution loses electron correlation effects as the Reviewer suggested, the investigation of the nature of the field-driven metallic state is outside the goals of this paper.

The Reviewer criticized the lack of many-body treatment for the quantum fluctuation of the gap parameter. As is the case with any static mean-field theory, our approximation would have overestimated the stability of the gap and the inclusion of the many-body effect will only make the switching field smaller, which would have further strengthened our main purpose of this work, that is, the quantum avalanche mechanism predicts unprecedentedly small switching fields.

We added the discussion below Eq.(8) in the revised manuscript.

We acknowledge the idea of quantum fluctuations to the gap parameter. We have considered that quantum fluctuations to the gap, such as the Goldstone modes, may also result in instability of the gap which we alluded to in the conclusion of the original manuscript. In fact, that was the initial idea we considered before we simplified it to phonons and came across the avalanche mechanism. We will pursue this idea in a future project.

> 4. What about the finite field case? Apart from shortcomings
> in handling the many body aspect, the assumption of a linear
> potential in the system is an oversimplification. The effect
> of a potential difference - introduced via leads - has been
> studied carefully and leads to a non trivial potential
> profile in the system, Riberio et al. Phys. Rev. B. 93, 144305
> (2016), Tanaka et al., Phys. Rev. B 83, 085113 (2011), Dutta
> et al., Phys Rev B 101, 245155 (2020). None of these seem to
> be known to the authors. As a result the entire spatial aspect
> of the breakdown process is lost in the present paper. This
> spatial aspect is not incidental, it suggests that the
> insulator-metal transition starts at the edge.
>
>

> Now on to (ii).
>

> 5. Only recently experimenters have started spatial mapping of the
> electronic state near a bias driven transition, see, e.g, Zhang et
> al., Phys Rev X 9, 011032 (2019). In Ca_2RuO_4 it shows via near
> field optical microscopy how the effect of a voltage bias propagates
> into the system. The bias driven metallisation
> process is not homogeneous.
>

We greatly appreciate the above comments and would like to reply to them

simultaneously since they both address the importance of the spatial pattern formation.

In the majority of the resistive switching literature, the dominant phenomenological discussions have centered on the lateral pattern formation (i.e., formation of filament along the field direction) and its relevance to the peculiar IV relations. In Ref [22] authored by two of us in 2018, we identified filament formation as the primary mechanism for the peculiar I-V characteristics in resistive switching measurements, and we did find that the switching fields are reduced by some fraction due to the added freedom to the system. We also stated (Figure 5 of [22]) that the new phase nucleates from the edge unless the impurities assist the switching. The reduction, however, is only a moderate fraction and cannot explain the many orders of magnitude discrepancy.

The Reviewer, however, seems to address a different pattern formation, with domains running perpendicular to the field. The references, Rebeiro et al, and Tanaka et al, are on one-dimensional models and Dutta et al (although two-dimensional) showed the switching as a function of bias rather than that of an electric field. For the reason that the majority of the resistive switching discussions are on the filament formation, we did not cite the works while we have been aware of the works. We acknowledge they are a valuable asset to the literature although there are uncertainties about the voltage profiles near contacts and the treatment often lack the diffusive effects in micron-long samples.

We are very much grateful to the Reviewer for bringing to our attention this very interesting work on Ca₂RuO₄ which we were not fully aware of. The work shows that the pattern formation is non-filamentary and shows nucleation of metallic domains running non-collinear with the current. The patterns are quite regular and sinusoidal, which indicates a sign of an electronic mechanism. It is still unclear where we can place the CaRuO system in the vast map of other resistive switching systems. The CaRuO system provides a counter-example to other systems such as Vanadium oxides, and they may well represent a different switching class. The works by Ribeira, Tanaka and Dutta will be important in understanding this new finding.

The issue can be framed into an experimental question of whether the resistive switching is a bulk or device/nanoscale phenomenon, or in other words, whether the switching is a function of the bias voltage or the electric field. The CDW literature seems to have settled on the bulk point of view, and their resistive transitions are widely viewed as a function of an electric field. In our opinion, the new findings on CaRuO shed important new light on resistive switching. However, it is outside the scope of this manuscript to resolve experimental differences exhibited in different materials. In the revised manuscript, we added a new paragraph (the third one in the introduction) appreciating the physics of Ca₂RuO₄ and defined the scope of this manuscript as a theory of nonequilibrium phase transition. We thank the Reviewer for making this manuscript more balanced.

> 6. The temperature dependence of the critical voltage (or field)
> in a wide variety of materials shows E_c reducing with increasing
> temperature. This ranges from the oxides - Ca₃Ru₂₀₇ and VO₂

> - to the organics kappa-BEDT and K-TCNQ. If the avalanche
> mechanism in Fig.3(g) were at play that would not have been
> the case. In fact in a correlated insulator the gap itself
> would reduce with T - weakening the threshold field - so it is
> hard to see where something like 3(g) is relevant.
>

Before we get to the specifics of the experimental issues, let us preface this discussion with our viewpoints. Our theoretical arguments rest on the assumption that the CDW and Mott systems, despite quite different energy scales and apparently different mechanisms, would share some common features in their respective resistive transition.

(a) The study of the threshold behavior in CDW materials has a long and rich literature. One of the well-established facts is the existence of low-field thresholds which increase with temperature, as summarized by Bardeen in his Physics Today article [1] (sometimes denoted as E_T' instead of E_T in experimental literature, for example [51]). Please note that this behavior occurs at quite a low temperature and bias so that the order parameter is unaffected, well away from the regime that the Reviewer discusses. The temperature dependence has been intensely debated, mainly within the classical disorder theory of Fukuyama-Lee-Rice model and Bardeen's Landau-Zener argument in quasi-1d limit, and there had been no fully accepted theories. Our avalanche mechanism reproduces the correct temperature dependence within a well-defined and an elementary model of Eq.(2) that provides an electronic/quantum switching mechanism which has been sought after for a long time.

(b) In the regime where the order parameter is reduced by temperature, as the Reviewer suggests, we have the decreasing switching field as shown in Fig. 3(g) in the high-T limit, and in Fig. 4(b) and 4(d) close to the transition temperature. We see that our results are fully consistent with the conventional view in this regime. In the revised manuscript, we make an experimental comparison by overlaying the measured $E_{\text{IMT}}(T)$ in VO₂ (red squares) to Fig. 4(b) and 4(d). It is remarkable that the theory correctly reproduces the concave I-V lineshape as well as the $E_{\text{IMT}}(T)$ as a decreasing function of temperature. We used the experimental transition temperature to scale the x-axis. For y-axis, we do not have the full temperature dependence and used an arbitrary scale.

(c) The above discussions in (a) and (b) demonstrate that our scenario is consistent with well-established experimental facts. Apart from it, by extending the belief that the CDW and Mott systems share common features, our theory makes a prediction that even in Mott-like systems the sub-gap quantum behavior of (a) would show up. Unfortunately, experimental data for resistive switching is scarce in the low-temperature limit. While more experimental work is required, we present in the revised manuscript results of IMT in TiS₃ system reported by two of the authors (cyan circles). Similar to the above, we rescaled the x-axis with experimental $T_c = 290$ K at the given gate voltage and used an arbitrary scaling for y-axis.

> 7. Apart from highlighting a scenario I am unable to see what
> existing discrepancy is actually explained, and data organised,

> by the authors. Given the note in which the introduction is written
> one would have hoped for an actual comparison of the predictions of
> the "avalanche approach" with breakdown fields and their temperature
> dependence in several materials.
>

We have addressed the Reviewer's concern in replying to previous comments,
and by comparing the experimental temperature dependence with our
theory.

>
> Let me summarise:

>
> The authors have introduced an interesting scenario that could play a
> role in bias/field driven transition in insulators. Electron-phonon
> coupling and a somewhat mysterious dephasing rate play a crucial role
> in the process - involving multiphonon emission and emergence of
> subgap states. This much is a conceptual advance and the effect could
> well play a role in real materials. However the treatment of actual
> correlated models is rather naive, some assumptions and predictions
> are clearly not consistent with experiments, and no effort is made to
> test out the predictions (whatever the methodological limits) against
> real data.

>
> I think the interesting idea should have a stronger foundation
> in theory and there should be an attempt at a serious comparison
> with experiments to merit publication in Nat Comm.

>
>
We thank Reviewer #2 for recognizing the novelty of our work and
making thoughtful criticism. We improved the presentation of our work,
and, in particular, supported our theory by direct incorporation of
the experimental data into the theoretical phase diagram. With the
clarifications provided here and the revisions in the manuscript, we
hope that the work merits publication in Nat Comm.

REVIEWER COMMENTS

Reviewer #1 (Remarks to the Author):

The manuscript has been improved suitably and now is ready for publication in Nature Communications. The authors have categorized the experimental facts, which can highlight the novelty of the quantum avalanche mechanism.

Reviewer #3 (Remarks to the Author):

I have read the two Referee reports and the response of the author to the Reviewer criticisms, especially those by Reviewer 2.

Reviewer 2 had remarked, "This is an innovative idea in an otherwise crowded and technically evolved field. It would be of value if (i) it is borne out by a credible many body calculation, and (ii) the predictions compare successfully with existing data. In my reading of the present version of the paper neither seems to be achieved."

In my opinion, the authors have gone some distance addressing the first issue, although not very satisfactorily. In Figure 2, plots of charge excitation as a function of the electric field do not, in my opinion, confirm or rule out the relation in Eq. 2 proposed by hand-waving arguments. Since Fig. 2 is obtained from actual calculation, I would like to see a plot of E_{av} extracted from these calculations plotted as a function of Γ , Δ and other phenomenological parameters that go into Eq. 2. At the moment, the calculations and Eq. 2 look disconnected to me.

Part (ii) of the Reviewer's criticism on experimental comparison appears addressed in Figure 4.

We appreciate the Reviewers for their helpful comments. Reviewer #1 is satisfied with the previous manuscript, and we only address Reviewer #3 in this reply. His/her comments are very clear and direct: Verify Eq. (2) with respect to various model parameters. While the manuscript was reviewed, we have been studying the very question with a new graduate student (who has been added as a co-author) and we report the results in the Supplementary Material.

To summarize, we verified that Eq. (2) correctly captures the tendency of the avalanche field with respect to all the parameters (except for the effective mass that sets the overall energy scale). Full many-body results fall within the reasonable range of agreement, with a general trend for systematic deviation at high avalanche fields. We justify the origin for the deviation as the nonperturbative effects in SM.

In the previous SM, we reported the behavior for a limited range of the phonon frequency and the current discussion is inclusive and replaces the last paragraph in SM.

We satisfactorily addressed the Reviewer's concerns and we hope that you will agree that our manuscript is now suitable for publication at Nature Communications.

>
> Reviewer #3 (Remarks to the Author):
>
> I have read the two Referee reports and the response of the author to
> the Reviewer criticisms, especially those by Reviewer 2.
>
> Reviewer 2 had remarked, "This is an innovative idea in an otherwise
> crowded and technically evolved field. It would be of value if (i) it
> is borne out by a credible many body calculation, and (ii) the
> predictions compare successfully with existing data. In my reading of
> the present version of the paper neither seems to be achieved."
>
> In my opinion, the authors have gone some distance addressing the
> first issue, although not very satisfactorily. In Figure 2, plots of
> charge excitation as a function of the electric field do not, in my
> opinion, confirm or rule out the relation in Eq. 2 proposed by
> hand-waving arguments. Since Fig. 2 is obtained from actual
> calculation, I would like to see a plot of E_{av} extracted from these
> calculations plotted as a function of Γ , Δ and other
> phenomenological parameters that go into Eq. 2. At the moment, the
> calculations and Eq. 2 look disconnected to me.
>

We greatly appreciate the Reviewer's criticism. As noted above, we have already been working on the issue and could confirm the validity of Eq.(2). As in figS1 in the Supplementary Material, the Log-Log plot of E_{av} vs. all model parameters shows the exponent one dependence in the low switching limit, which is in agreement with Eq (2) except for the phonon frequency. The only notable disagreement is the exponent 3/2 to the phonon frequency. We note, however, that the expression for the electron-phonon coupling used to derive Eq (2) is very crude in that the

phonon frequency is unreliably factored out of an integral of electron self-energy. As mentioned in the previous manuscript, the dependence on the fermion damping is the most nontrivial and the most numerically reliable. The super-linear deviation by dephasing rate is due to the interaction-generated dephasing, and the sub-linear of el-ph parameters is due to the enhancement of the nonperturbative coupling effects. Therefore, we believe that within the schematic nature of Eq (2) the hand-wavy relation is well-justified.

> Part (ii) of the Reviewer's criticism on experimental comparison
> appears addressed in Figure 4.
>
>

REVIEWERS' COMMENTS

Reviewer #3 (Remarks to the Author):

The authors have reasonably addressed my question on checking the consistency of Eq. 2 (obtained by hand-waving arguments) with the numerical calculations. The numerical results show linear scaling with the damping parameter Γ , while the scaling of other parameters such as g_{ep} and ω_{ph} look rather different from Eq. 2. However I am happy to recommend the paper for publication.

Dear Editor,

Please find our thoughts to Reviewer #3's comments below.
We hope that we have addressed all concerns of the reviewers.

>
> REVIEWERS' COMMENTS
>
> Reviewer #3 (Remarks to the Author):
>
> The authors have reasonably addressed my question on checking the
> consistency of Eq. 2 (obtained by hand-waving arguments) with the
> numerical calculations. The numerical results show linear scaling with
> the damping parameter Γ , while the scaling of other parameters such
> as g_{ep} and ω_{ph} look rather different from Eq. 2. However I am
> happy to recommend the paper for publication.
>

We thank Reviewer #3 for understanding the limitations of the heuristic derivation. In self-consistent diagrammatic calculations, strong el-ph coupling is known to lead to correlated states (such as localized polarons etc.) and such nonlinear contributions exist in the calculation even though we are away from such extreme regimes in the reported results. Even with moderate nonlinear effects, it is expected that the heuristic derivation, Eq. (2), will deviate from the full results. As mentioned in the Supplementary Info, nonlinear effects effectively enhances the el-ph coupling and lowers the avalanche field to the sublinear deviation.

Aside from such details, the fact that the avalanche field is an increasing functions of the parameter plotted in FIG.S1 shows without much doubt that the argument leading to Eq (2) "is conceptually sound and captures the essential physics" as stated in the SI.